# Deliberative panels as a source of public knowledge: A large-sample test of the Citizens' Initiative Review

**John Gastil**[1]*, **Kristinn Már Ársælsson**[2], **Katherine R. Knobloch**[3], **David L. Brinker**[4], **Robert C. Richards, Jr**[5], **Justin Reedy**[6], **Stephanie Burkhalter**[7]

1 Department of Communication Arts & Sciences, Department of Political Science and School of Public Policy, Pennsylvania State University, University Park, Pennsylvania, United States of America, 2 Behavioral Science, Duke Kunshan University, Kunshan, Jiangsu, China, 3 Department of Communication, Colorado State University, Fort Collins, Colorado, United States of America, 4 Brinker Consulting, Concord, Massachusetts, United States of America, 5 Clinton School of Public Service, University of Arkansas, Little Rock, Arkansas, United States of America, 6 Department of Communication, University of Oklahoma, Norman, Oklahoma, United States of America, 7 Department of Politics, California State Polytechnic University, Humboldt, California, United States of America

* jgastil@psu.edu

**Data Availability Statement:** The data and code are available on Harvard Dataverse (https://doi.org/10.7910/DVN/IRVPKN).

## Abstract

Evolving US media and political systems, coupled with escalating misinformation campaigns, have left the public divided over objective facts featured in policy debates. The public also has lost much of its confidence in the institutions designed to adjudicate those epistemic debates. To counter this threat, civic entrepreneurs have devised institutional reforms to revitalize democratic policymaking. One promising intervention is the Citizens' Initiative Review (CIR), which has been adopted into law in Oregon and tested in several other states, as well as Switzerland and Finland. Each CIR gathers a demographically stratified random sample of registered voters to form a deliberative panel, which hears from pro and con advocates and neutral experts while assessing the merits of a ballot measure. After four-to-five days of deliberation, each CIR writes an issue guide for voters that identifies key factual findings, along with the most important pro and con arguments. This study pools the results of survey experiments conducted on thirteen CIRs held from 2010 to 2018, resulting in a dataset that includes 67,120 knowledge scores collected from 10,872 registered voters exposed to 82 empirical claims. Analysis shows that reading the CIR guide had a positive effect on voters' policy knowledge, with stronger effects for those holding greater faith in deliberation. We found little evidence of directional motivated reasoning but some evidence that reading the CIR statement can spark an accuracy motivation. Overall, the main results show how trust in peer deliberation provides one path out of the maze of misinformation shaping voter decisions during elections.

## Introduction

Every year, referenda and initiatives ask voters to evaluate complex public policy questions that appear on their ballots. This "demanding form of democracy" [1] has become more

**Funding:** Gastil, J. (2015). Principal Investigator, The Democracy Fund. "2015-2016 Citizens' Initiative Review Study and Reporting." ($75,000) https://democracyfund.org/ Gastil, J., & Knobloch, K. (2014). Co-Principal Investigators, National Science Foundation (Directorate for Social, Behavioral and Economic Sciences: Decision, Risk and Management Sciences, NSF Award #1357276/1357444). "Collaborative research: A multi-state investigation of small group and mass public decision making on fiscal and scientific controversies through the Citizens' Initiative Review." ($418,000) https://www.nsf.gov/ Gastil, J. (2013). Pennsylvania State University Social Science Research Institute. Award for summer workshop bringing together researchers investigating the Oregon Citizens' Initiative Review ($5,000) https://ssri.psu.edu/ Gastil, J., & Knobloch, K. (2012). Joint learning agreement (research contract) with the Kettering Foundation, with 76% of the budget allocated to Pennsylvania State University and 24% to Colorado State University. "Examining deliberation and the cultivation of public engagement at the 2012 Oregon Citizens' Initiative Review" ($30,000) https://www.kettering.org/ Gastil, J. (2010). Principal Investigator, National Science Foundation (Directorate for Social, Behavioral and Economic Sciences: Decision, Risk and Management Sciences and Political Science Programs, NSF Award # 0961774), "Investigating the Electoral Impact and Deliberation of the Oregon Citizens' Initiative Review" ($218,000) https://www.nsf.gov/ Gastil, J. (2010). Principal Investigator, University of Washington Royalty Research Fund. "Panel Survey Investigation of the Oregon Citizen Initiative Review" ($40,000) https://www.washington.edu/research/or/royalty-research-fund-rrf/ The funders had no role in study design, data collection and analysis, decision to publish, or preparation of the manuscript.

**Competing interests:** The authors have declared that no competing interests exist.

daunting as the public information environment suffers from what the RAND Corporation calls an epidemic of "truth decay" [2]. Namely, the evolving US media and political systems have left the public divided over objective facts and have diminished faith in the institutions designed to adjudicate those debates. A recent essay in *Science* advocated countering these threats by developing a rigorous "science of deliberation" to assess institutional reforms that might revitalize democratic policymaking [3]. Defined therein as a blend of "argumentative complexity" and "civility," deliberation hinges on voters accessing relevant "facts and evidence" to aid them in reaching well-informed judgments.

One of the interventions cited in that essay was the Citizens' Initiative Review (CIR), which has been adopted into law in Oregon and tested in several other states, as well as Switzerland and Finland [4]. In an era of declining trust in government and public institutions [5], the CIR offers skeptical voters a "trusted information proxy." The CIR process was created by experienced practitioners of deliberation [4,6]. In the State of Oregon, which has held the most panels, a CIR Commission selects ballot measures for review, then contracts with a nongovernmental organization to create a CIR panel from a stratified random sample of 20–24 registered voters [7]. The CIR panel engages in question-and-answer sessions with pro and con advocates and issue experts over 4–5 days, along with extensive small group discussion on the ballot measure. The experts are individuals with issue-relevant knowledge working at public agencies, nongovernmental organizations, or universities. The process culminates in the panel writing a one-page analysis that highlights key findings of fact and what the panel considers as the strongest, factually accurate arguments for and against a ballot measure. The CIR's analysis appears as a "Citizens' Statement" in the official *Voters' Pamphlet* mailed to every registered voter in Oregon.

In theory, the CIR should work as an effective means of education on policy questions for which mistaken public beliefs have yet to become ideological convictions [8]. When tasked with legislating via initiative and referendum elections, voters often lack basic policy knowledge and lean on preexisting partisan biases [1,9]. To improve voter knowledge, the CIR benefits from a source credibility akin to the public's trust in the jury system [10,11] and randomly-selected representative bodies more generally [12–14]. The CIR leverages that credibility to provide simplified, timely information to voters [15], who can be daunted by the cognitive demands of voting on legislation ([16–18]. A "science of deliberation," however, requires a reliable assessment of the net impact of the CIR on public knowledge, and this study provides a firm test of the hypothesis that reading a CIR Citizens' Statement increases voter knowledge.

## Scope of research

This study combines results from survey experiments on thirteen CIRs held from 2010 to 2018 in Oregon, Arizona, California, Colorado, and Massachusetts. Participants were recruited through a Qualtrics online panel, though one mail survey (2014) and one online survey (2012) used a publicly-available list of registered voters. Researchers (or Qualtrics) had initial contact information for respondents, but these personally identifying contact lists were unlinked to survey responses and destroyed after data collection. We received exemption determinations permitting the use of implied consent for mail, phone, and internet voter surveys from the University of Washington Human Subjects Division and the Pennsylvania State University Office for Research Protections.

Each survey was conducted in the final month before the election, and those who had already seen the official voter guide and/or CIR statement were excluded from analysis. These surveys, principally funded by the National Science Foundation, tested whether the CIR improved the accuracy of voters' policy beliefs by (a) randomly varying the information

exposure that voters received in advance of an election then (b) measuring their knowledge on a series of pertinent true/false questions. On each knowledge item, respondents received a Factual Accuracy score ranging from -2 (confident in an incorrect answer) to +2 (confident in a correct answer) to see whether the CIR improved the accuracy of voters' policy beliefs [19,20].

Pooling all the surveys into one dataset, we used multilevel ordered logit regressions using odds ratios (OR) to analyze a total of 67,120 Factual Accuracy scores calculated for 10,872 registered voters. This yields statistical power sufficient to detect even very small effect sizes [21]. Such multi-year datasets covering several policy issues are rare, and they provide more reliable and precise estimation of main effects of deliberative processes. (See Online Supplement S1 Table in S1 File for details about the surveys combined in this dataset.)

## Findings

### Overall impact on public knowledge

Across the thirteen CIRs, reading the CIR statement increased voters' knowledge about a ballot measure ($OR$ = 1.45, $SE$ = .06, $p$ < .001): 44% of those who were *not* shown the CIR statement (control) picked the correct true/false option, compared to about 60% of those who saw it (treatment). Relative to the control group, exposure to the CIR statement reduced the number of incorrect responses by about 8 percentage points (27% decrease) and increased the number of correct answers by about 10 points (23% increase). The CIR's largest boost was in the proportion of respondents who were confident in their knowledge when their belief was accurate. Fig 1 shows the knowledge gains after controlling for different ballot measures and sample sizes (See S2-S4 Tables and S1 Fig in S1 File).

The surveys included in this analysis addressed thirteen policy questions across five US states. Fig 2 shows that the CIR guide improved average Factual Accuracy for all but one of those issues. Only the inaugural CIR—concerning mandatory minimum sentencing in Oregon—generated no improvement. Significant effect sizes (Cohen's $d$) ranged between .10 and .40 with an overall average effect of .18 ($CI$ = .12 .24, (see Online Supplement S2 Fig in S1 File).

Another way of assessing the consistency of the CIR's effect on Factual Accuracy breaks the results down by individual factual claims. Each of the thirteen policy questions had a corresponding set of true/false statements, resulting in a total pool of 82 knowledge items. Among those knowledge items, 69 had sufficient respondents to detect at least an effect if $d$ > .4 with a power of 80%. Using t-tests and a two-tailed .05 significance threshold, 33 of these 69 items had significant positive effects (mean $d$ = .37), 33 had non-significant effects (mean $d$ = .09), and three of them had significant *negative* effects (mean $d$ = -.30). After applying Holm multiple hypothesis correction, 21 items with a positive effect were statistically significant, none with negative effects, and 48 not statistically significant (see Online Supplement S5 Table in S1 File). Simply put, the CIR had a strong tendency to boost public knowledge, though in most cases its effect was small. It is not, however, foolproof: In less than five percent of cases, reading the statement backfired by significantly reducing the accuracy of public beliefs. (See Stata log files on request and replication materials for full details).

This first analysis showed the CIR's overall benefit relative to either no information guide *or* an official voter guide. Official guides are commonly written by state officials, such as the Secretary of State, and their descriptions of ballot measures often use technical language constrained by neutrality concerns to the detriment of direct policy relevance [15]. Thus, we tested whether those who only read the CIR statement had Factual Accuracy scores above those who only read the official guide. (The subset of CIR issues available for this analysis included: sentencing, casinos, and corporate taxes in Oregon; housing in Portland Metro and California; and nursing in Massachusetts.) We found that they do

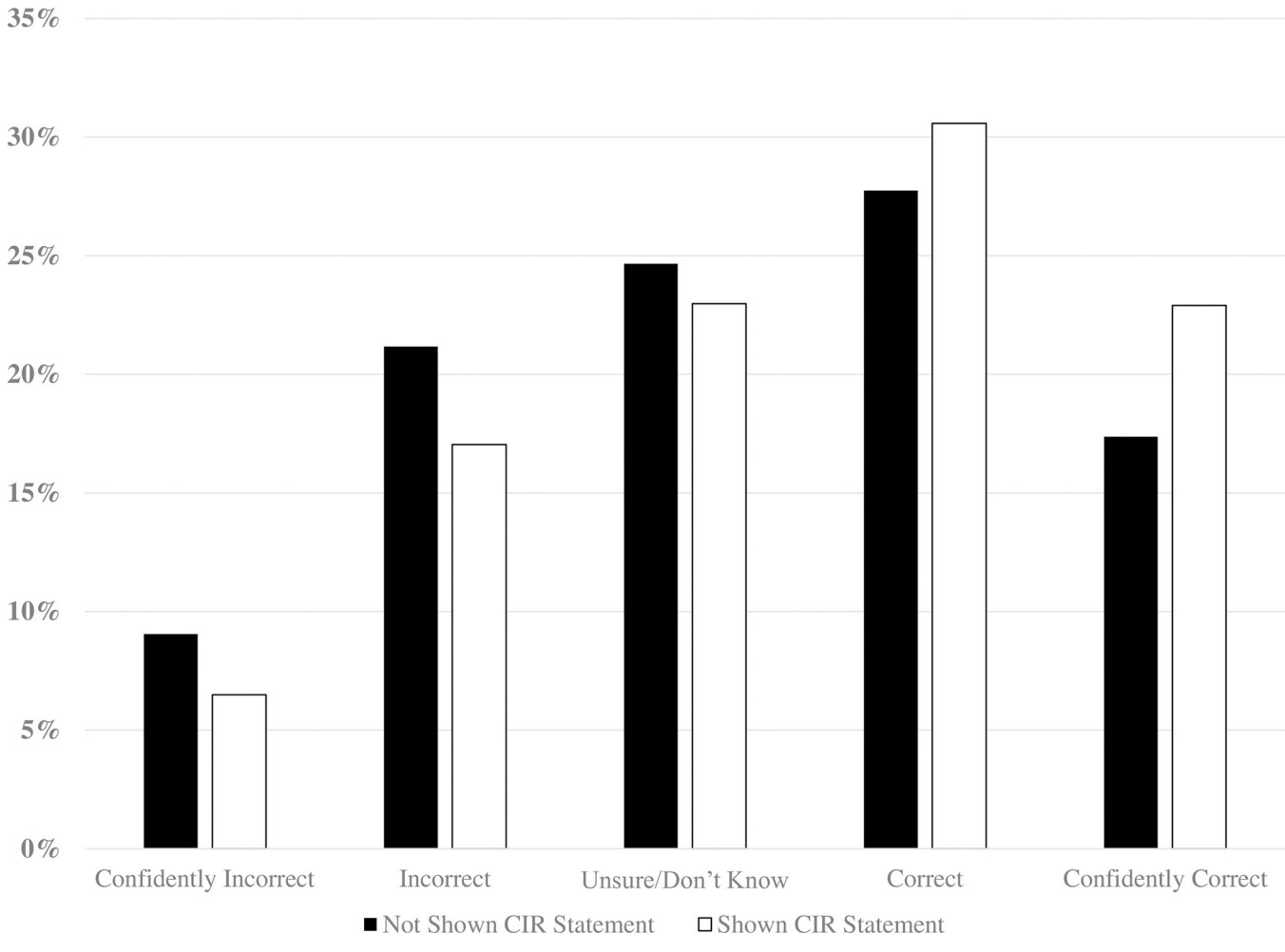

**Fig 1. Policy knowledge gains from exposure to a CIR statement.**

($OR$ = 1.32, $SE$ = .14, $p$ = .006), although the difference is more modest. About 6 percent more of respondents pick the correct answer on individual knowledge items—50% compared to 44%—if they read the CIR statement instead of the official voter guide (see Online Supplement S6 Table and S3 Fig in S1 File).

We then examined whether reading the CIR statement *adds* knowledge gains when presented alongside the official voting guide, compared to those who read only the voting guide. (The CIR issues available for this analysis included: corporate taxes in Oregon, housing in Portland Metro; marijuana in Oregon and Arizona; and nursing in Massachusetts.) When presented together in this way, exposure to the CIR still has a modest net benefit ($OR$ = 1.62, $SE$ = .12, $p$ < .001). Fig 3 shows that on average, exposure to the CIR statement *and* the voting guide produced significantly larger knowledge gains than reading only the official guide. About 60% picked the correct answer if they read the CIR statement and voting guide compared to 49% among those who read the voting guide without the CIR statement (see Online Supplement S7 Table in S1 File). We also estimated the effect of reading the CIR compared to no exposure and found a similar effect ($OR$ = 1.58, $SE$ = .15, $p$ < .001), as shown in S8 Table in S1 File.

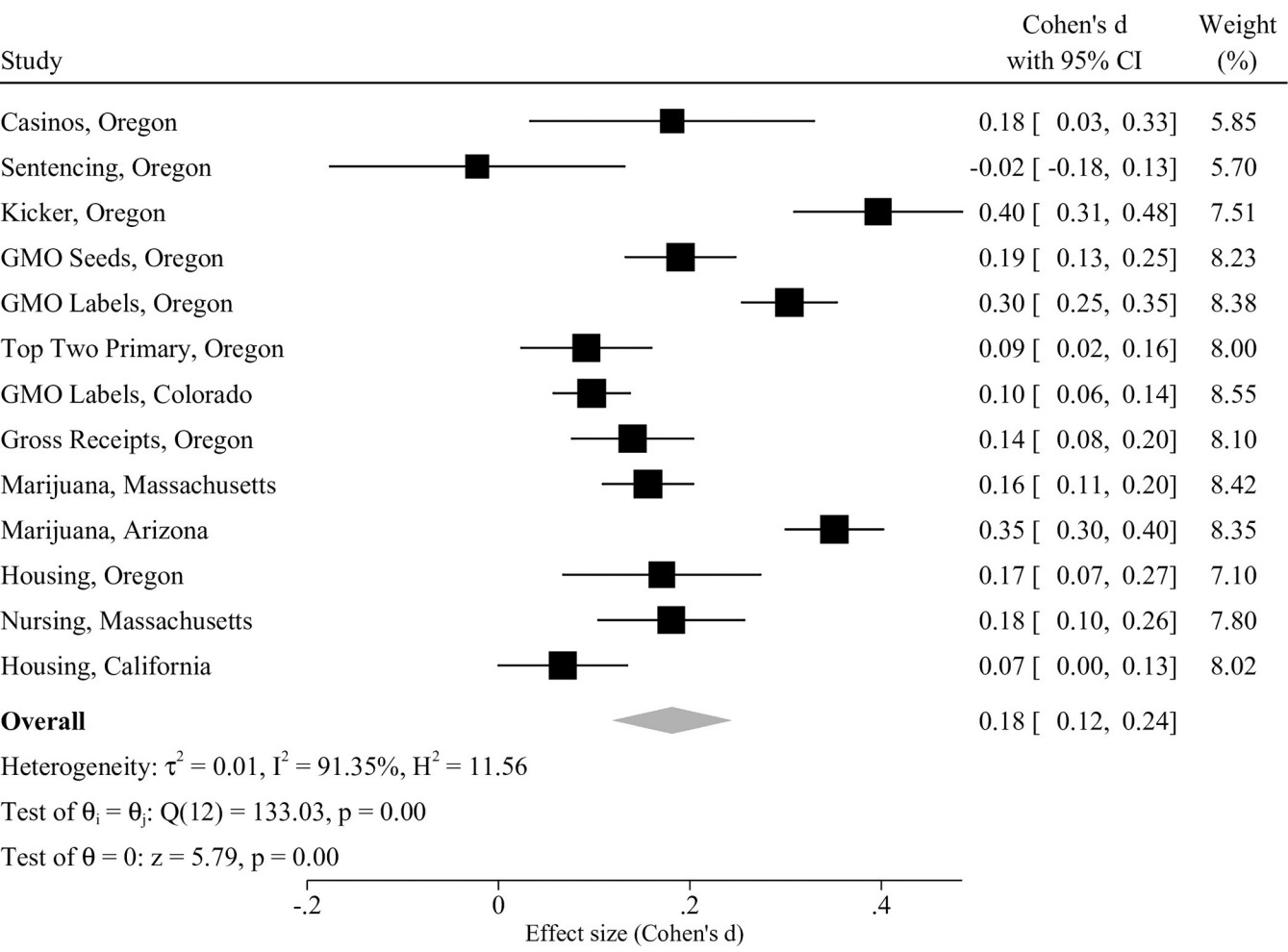

**Fig 2. Internal meta-analysis of CIR effect sizes on factual accuracy.**

## Partisan bias versus accuracy motivation

The CIR model stands on the assumption that voters reading a nonpartisan issue statement are motivated to understand the issues before them [22]. By contrast, motivated reasoning theory and cultural cognitive theory posit that partisans reject information infelicitous to their prior biases [23]. The latter view cautions that observed knowledge gains might reflect shifts in empirical beliefs among only those already predisposed to agree with the claims made in a CIR statement. Those who initially held inaccurate beliefs that buttressed their identities would reject any contradictory information provided by the CIR. Directional motivated reasoning has repeatedly been reported among Democrats and Republicans [2,19,20,22,23].

If our findings above are driven by directional motivated reasoning, we expect the partisan group who was more factually accurate pre-treatment to improve more; and those less accurate to ignore the summary provided by the minipublic. Alternatively, Democrats and Republicans improve their Factual Accuracy at a similar rate or the minipublic statement induces an accuracy motivation [22], where those less factually accurate pre-treatments improve more after reading the statement.

Testing this requires two steps. First, we need to estimate whether there is a significant difference in the control group (who were not exposed to any treatment) between Democrats and

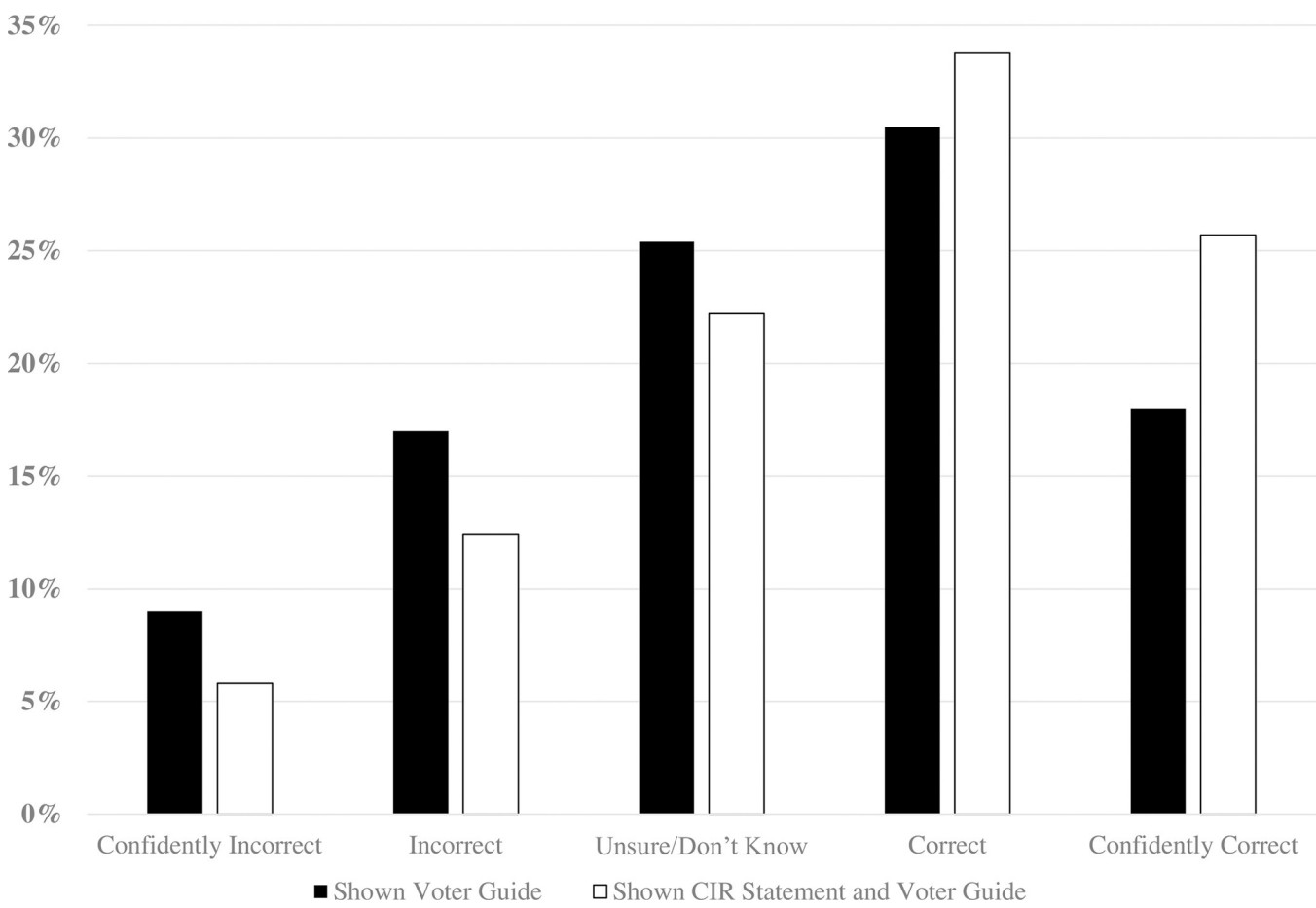

**Fig 3. Policy knowledge gains from exposure to the CIR statement and voter guide compared to the voter guide only.**

Republicans in Factual Accuracy. Next, we test whether there is a significant difference in treatment effect across Democratic and Republican study participants or they improve at a comparable rate. Then, we employ a two-pronged analysis. First, we used our pooled dataset. With 33,614 claim responses we can detect effects with $d > .025$ with a power of 99%. In the control group we find that Democrats ($M = .18$, $SE = .01$) compared to Republicans ($M = .13$, $SE = .01$) are marginally more Factually Accurate, $t(22,171) = 3.3$, $p = .001$. However, multi-level regression analysis suggests treatment effects are comparable across the two partisan groups ($OR = 1.03$, $SE = .10$, $p = .79$, see Online Supplement S9 and S10 Tables in S1 File).

Our analysis using pooled data might produce biased results if the knowledge gains are split across different claims. Thus, our second approach tested Knowledge Items where we find a significant difference in Factual Accuracy. For this analysis we used eighteen highly powered true/false claims across two CIR statements (GMO Labels in Oregon and Colorado). In total there are 82 true/false claims in our dataset. In this analysis we exclude under-powered studies ($N < 875$). We can detect negligible effects $d > .10$ with 90% power. The results are comparable if we run the analysis for all 82 true/false claims and for all treatment regimes.

After applying Holm multiple hypothesis correction, we found three claims where Democrats were significantly more Factually Accurate in the control group. In one of these instances, the average treatment effect was significantly ($p = .019$) higher among Republicans, suggesting accuracy motivation, but only before Holm correction ($p = .32$).

Taken together, the CIR statement had a similar effect on both Democrats and Republicans. In rare cases, the group that began with lower Factual Accuracy improved its scores more than its counterpart, and we found no evidence of the opposite. Thus, an accuracy motivation appeared to be at work more than a directional one. (See Stata log files on request and replication materials for full details.)

## Confidence in deliberation and the CIR process

Our final analysis tested a core assumption of the CIR. Theoretically, much of the CIR's influence comes from the confidence that busy voters place in a deliberative body of their peers, which can serve as a "trusted information proxy" [6]. If this holds true, those respondents who expressed more confidence in public deliberation should show stronger information gains.

To scrutinize that assumption, three survey items were combined into a scale measuring respondents' faith in citizen deliberation ($\alpha$ = .66). The three items read: "Even people who strongly disagree can make sound decisions if they sit down and talk;" "everyday people from different parties can have very civil conversations about politics;" and "the first step in solving our common problems is to discuss them together." Higher Factual Accuracy gains from reading the CIR statement came for those who had confidence in the ability of "everyday people" to solve "common problems" through deliberation ($OR$ = .1.22, $SE$ = .05, $p <$ .001). Thus, the CIR's impact was *partly* a matter of trusting in the citizen deliberation on which it relies. (For more detail, see S11 Table in S1 File)

## Discussion

Given that "deliberative institutional experimentation is flourishing throughout the world" [3] more rigorous investigation of democratic reforms is important. Databases such as Participedia.net provide a catalogue of thousands of case studies of deliberation, but aggregated analyses of such cases often lack precision, a consistent set of variables, comparable measures of key variables, and sufficient statistical power [24,25]. Such repositories also risk selection bias, owing to the tendency to underreport unsuccessful cases [26].

Our analysis of the CIR overcame those obstacles by including *every* instance of this deliberative process, using largely overlapping sets of variables, including a true/false knowledge test that remained consistent in its structure across cases but was tailored to each unique policy question. The result was a database of responses from tens of thousands of respondents, including close to seventy thousand knowledge item responses. Combining the raw data from each survey has permitted the first fine-grained analysis of a deliberative minipublic's wider impact on an electorate over several iterations.

The two principal conclusions these data afford are these. First, the CIR model *works* as a means of improving public knowledge in advance of elections. This method of public education also outperforms conventional voter guides produced by state officials. When reading the CIR, voter knowledge improved regardless of prior ideological biases. If anything, voters more often *corrected* their ideological biases, rather than rejecting information that conflicted with prior beliefs. The CIR is no panacea, but one can imagine many applications to harness the power of smaller deliberative bodies to inform the judgment of the wider public, perhaps even beyond the confines of elections [14,27].

Second, confidence in the value of public deliberation enhances knowledge gains by priming individuals to receive the wisdom generated by a deliberative body of their peers. The trust effect was modest, however, which suggests that the CIR's efficacy as a source of public information does not hinge on people's faith in deliberation.

Future research on voting guides like the CIR could track voter knowledge and attitudes longitudinally, with pre- and post-exposure measures of empirical beliefs. This would make it possible to trace, at the individual level, how preexisting attitudes toward deliberation shape the movement from fiction to fact and from uncertainty to confident accuracy. This would clarify the degree to which a trusted deliberative body like the CIR is equally adept at correcting misperceptions as it is in building voters' confidence in accurate beliefs they already hold.

Caveats aside, this study shows that small scale deliberative interventions can lead to clear knowledge gains. Since over forty percent of Oregon voters seek out the CIR on their own when it appears in the official *Voters' Pamphlet* [4], this suggests a widespread knowledge increase in the electorate. We hope that these results inspire future systematic, aggregated analyses of other deliberative bodies, such as larger Citizens' Assemblies, Deliberative Polls, Citizens' Juries, and other "minipublics" that harness the power of small deliberative bodies for larger public purposes [28–30]. The movement toward more consistent evaluation of such processes points in the right direction [31], and we hope that our study becomes one of many effective efforts to advance the science of deliberation. In a world beset by misinformation and misperception [2], such a body of knowledge could not be more timely.

## Supporting information

**S1 File. Online supplement: Tables and figures.**
(DOCX)

## Acknowledgments

The authors thank all of those who have made possible this ongoing program of research, including our wider team of collaborators noted at the CIR Research Project site (https://sites.psu.edu/citizensinitiativereview) and Healthy Democracy, which provided open access to the CIR process itself.

## Author Contributions

**Conceptualization:** John Gastil, Kristinn Már Ársælsson, Katherine R. Knobloch.

**Data curation:** John Gastil, Katherine R. Knobloch, David L. Brinker, Robert C. Richards, Jr.

**Formal analysis:** John Gastil, Kristinn Már Ársælsson.

**Funding acquisition:** John Gastil, Katherine R. Knobloch.

**Investigation:** John Gastil, Katherine R. Knobloch, David L. Brinker, Robert C. Richards, Jr, Justin Reedy, Stephanie Burkhalter.

**Methodology:** John Gastil, Kristinn Már Ársælsson, Katherine R. Knobloch, David L. Brinker, Robert C. Richards, Jr, Justin Reedy.

**Project administration:** John Gastil.

**Resources:** John Gastil.

**Writing – original draft:** John Gastil, Kristinn Már Ársælsson.

**Writing – review & editing:** John Gastil, Kristinn Már Ársælsson, Katherine R. Knobloch, David L. Brinker, Robert C. Richards, Jr, Justin Reedy, Stephanie Burkhalter.

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
