## [Decision Letter · Decision Letter 0]

26 Apr 2023

PONE-D-23-06764Deliberative panels as a source of public knowledge: A large-sample test of the Citizens’ Initiative ReviewPLOS ONE

Dear Dr. Gastil,

Thank you for submitting your manuscript to PLOS ONE. After careful consideration, we feel that it has merit but does not fully meet PLOS ONE’s publication criteria as it currently stands. Therefore, we invite you to submit a revised version of the manuscript that addresses the points raised during the review process.

 I encourage you to consider the reviewers' comments. Moreover, I have the following queries:-P. 68, you refer to randomly-selected representative bodies. But my understand is that it these samples are not random and include only registered voters. Correct? If so, please clarify. -You refer to experts that will inform citizens. Can you please provide some information about the kind of people that they are. I am not calling for a discussion about the definition of an 'expert' but simply some descriptive information. -On the 82 tests when you unpack the knowledge items. Why did you not correct for multiple hypothesis testing?

We look forward to receiving your revised manuscript.

Kind regards,

Jean-François Daoust

Academic Editor

PLOS ONE

Journal Requirements:

 "Gastil, J. (2015). Principal Investigator, The Democracy Fund. “2015-2016 Citizens' Initiative Review Study and Reporting.” ($75,000) https://democracyfund.org/

Gastil, J., & Knobloch, K. (2014). Co-Principal Investigators, National Science Foundation (Directorate for Social, Behavioral and Economic Sciences: Decision, Risk and Management Sciences, NSF Award #1357276/1357444). “Collaborative research: A multi-state investigation of small group and mass public decision making on fiscal and scientific controversies through the Citizens’ Initiative Review.” ($418,000) https://www.nsf.gov/

Gastil, J. (2013). Pennsylvania State University Social Science Research Institute. Award for summer workshop bringing together researchers investigating the Oregon Citizens’ Initiative Review ($5,000) https://ssri.psu.edu/

Gastil, J., & Knobloch, K. (2012). Joint learning agreement (research contract) with the Kettering Foundation, with 76% of the budget allocated to Pennsylvania State University and 24% to Colorado State University. “Examining deliberation and the cultivation of public engagement at the 2012 Oregon Citizens’ Initiative Review” ($30,000) https://www.kettering.org/

Gastil, J. (2010). Principal Investigator, National Science Foundation (Directorate for Social, Behavioral and Economic Sciences: Decision, Risk and Management Sciences and Political Science Programs, NSF Award # 0961774), “Investigating the Electoral Impact and Deliberation of the Oregon Citizens’ Initiative Review” ($218,000) https://www.nsf.gov/

Gastil, J. (2010). Principal Investigator, University of Washington Royalty Research Fund. “Panel Survey Investigation of the Oregon Citizen Initiative Review” ($40,000) " ext-link-type="uri" xlink:type="simple">https://www.washington.edu/research/or/royalty-research-fund-rrf/"

"The authors thank all of those who have made possible this ongoing program of research, including our wider team of collaborators noted at the CIR Research Project site (https://sites.psu.edu/citizensinitiativereview) and Healthy Democracy, which provided open access to the CIR process itself. Funding was made possible by The Democracy Fund (contract “2015-2016 Citizens' Initiative Review Study and Reporting”), the National Science Foundation (Directorate for Social, Behavioral and Economic Sciences: Decision, Risk and Management Sciences, Award #1357276/1357444 and Award # 0961774), a Kettering Foundation joint learning agreement (“Examining deliberation and the cultivation of public engagement at the 2012 Oregon Citizens’ Initiative Review”), and a University of Washington Royalty Research Fund grant (“Panel Survey Investigation of the Oregon Citizen Initiative Review”)."

"Gastil, J. (2015). Principal Investigator, The Democracy Fund. “2015-2016 Citizens' Initiative Review Study and Reporting.” ($75,000) https://democracyfund.org/

Gastil, J., Knobloch, K. (2014). Co-Principal Investigators, National Science Foundation (Directorate for Social, Behavioral and Economic Sciences: Decision, Risk and Management Sciences, NSF Award #1357276/1357444). “Collaborative research: A multi-state investigation of small group and mass public decision making on fiscal and scientific controversies through the Citizens’ Initiative Review.” ($418,000) https://www.nsf.gov/

Gastil, J. (2013). Pennsylvania State University Social Science Research Institute. Award for summer workshop bringing together researchers investigating the Oregon Citizens’ Initiative Review ($5,000) https://ssri.psu.edu/

Gastil, J., Knobloch, K. (2012). Joint learning agreement (research contract) with the Kettering Foundation, with 76% of the budget allocated to Pennsylvania State University and 24% to Colorado State University. “Examining deliberation and the cultivation of public engagement at the 2012 Oregon Citizens’ Initiative Review” ($30,000) https://www.kettering.org/

Gastil, J. (2010). Principal Investigator, National Science Foundation (Directorate for Social, Behavioral and Economic Sciences: Decision, Risk and Management Sciences and Political Science Programs, NSF Award # 0961774), “Investigating the Electoral Impact and Deliberation of the Oregon Citizens’ Initiative Review” ($218,000) https://www.nsf.gov/

Gastil, J. (2010). Principal Investigator, University of Washington Royalty Research Fund. “Panel Survey Investigation of the Oregon Citizen Initiative Review” ($40,000) " ext-link-type="uri" xlink:type="simple">https://www.washington.edu/research/or/royalty-research-fund-rrf/"

Reviewers' comments:

Reviewer's Responses to Questions

**Comments to the Author**

1. Is the manuscript technically sound, and do the data support the conclusions?

Reviewer #1: Yes

Reviewer #2: Yes

2. Has the statistical analysis been performed appropriately and rigorously? 

Reviewer #1: Yes

Reviewer #2: Yes

3. Have the authors made all data underlying the findings in their manuscript fully available?

Reviewer #1: Yes

Reviewer #2: Yes

4. Is the manuscript presented in an intelligible fashion and written in standard English?

Reviewer #1: Yes

Reviewer #2: Yes

5. Review Comments to the Author

Reviewer #1: This manuscript reports on research on Citizens’ Initiative Reviews (CIRs)—a process that attaches a citizens’ jury to ballot initiatives, with the aim of informing the public in advance of voting. This innovation addresses the problem that citizens are often quite ignorant of the issues upon which they are called to vote. The hope is that, with a modest investment, governments can improve the quality of citizens’ decision-making. The citizens’ jury in these cases is comprised of 20-24 citizens chosen through stratified random sampling so as to roughly represent the population of the relevant jurisdiction. The body learns about the issue on the ballot, hears from experts and advocates, deliberates, and then crafts a state with advice for distribution to voters. This research followed 13 CIRs using survey experiments to see whether the CIR statements increased voters’ knowledge. They find that, in almost all cases, there were modest but consistent and measurable increases in knowledge.

Overall, this manuscript is very tightly conceived, well-written, and convincingly researched. The findings are important for upgrading ballot measures, which are not just common, but can also go badly wrong if voters make poor or uninformed choices. Brexit is a case in point: polls show that had even a few voters known more about what they were voting for, the referendum would not have passed.

The manuscript really does not need revisions, but I have a three questions that might be very briefly addressed.

First, do the sampling periods correspond to the periods before the actual ballot measures? Are respondents asked about their exposure to, say, voter guides that contain the statements used in the survey?

Second, the question of respondents’ “confidence in the deliberative process and the CIR is a bit ambiguous (177ff). One can have confidence that “ordinary citizens” will learn and make good judgments. Or one can have confidence in the deliberative processes. The first bit of information is easier for citizens to gain; the second requires a bit more effort.

Third, the statement that this survey dataset “has permitted a fine-grained analysis not seen before in this field of study” is quite right for CIR processes. However, there is at least one other such dataset in the broader field of democratic innovations, that collected by Jim Fishkin and his colleagues on Deliberative Polling, which also involves a broad range of consistent questions aggregated over many dozens of events, all of which have close to the same structure.

Reviewer #2: This is a thorough and well-written analysis of how Citizens’ Initiative Reviews (CIRs) can increase citizens’ knowledge of policy issues prior to voting in a ballot initiative. The authors have a large and high quality dataset which they analyze effectively. This article makes an important contribution to the literature and will, in my opinion, be well cited. I have only a few smaller points that can be addressed through a minor review.

Theory/ framing/ concepts

- From the outset, the article is framed around the problem of growing polarization and misinformation. However, is polarization really the problem that CIRs address? It seems more like the problem CIRs address is a lack of awareness/ lack of opinion (not the problem of strongly held, polarized, partisan beliefs). If the problem is polarization, wouldn’t the solution be moving people toward some kind of more ambivalent position? Learning about the CIR increases both knowledge and confidence of one’s knowledge, which seems like the opposite of ambivalence. I am not saying that increasing confidence of one’s knowledge is a bad thing (increasing confidence of correct knowledge is, as the author’s point out, a good thing). Rather, I think this framing is simply mapping a solution to the wrong problem. This is a more classical epistemic problem of democracy: most people aren’t policy experts/ don’t know much about politics (and have little confidence in what they do know). The CIRs help people overcome this knowledge deficit so that they can make more informed (and confident) decisions about the things that affect them.

- Note that the authors talk about deliberation throughout the paper. It’s even a key moderating variable in the analysis. But deliberation is never defined for the reader. This term should be defined the first time it’s used. Also, more clarity on how respondents’ “faith in deliberation” is measured would be beneficial.

- I find it incredibly interesting that “the CIR’s largest boost was in the proportion of respondents who were confident in their knowledge when their belief was accurate.” This is a novel and important finding. But theoretically, it does not seem to me that increasing one’s confidence in one’s knowledge is the same as increasing knowledge itself. But here knowledge and confidence in knowledge are measured with a single outcome. I could be convinced that this is fine, but I’d like a little more background/theoretical justification for capturing what (at least at first blush) seem to be two concepts with a single outcome. What I'm saying is that a couple of sentences explaining the outcome would improve the paper (I'm not saying the analysis needs to be fundamentally changed in any way).

- P. 10: "In conclusion, this study shows that small scale deliberative interventions can lead to widespread knowledge gains." I'm not sure this conclusion can quite be drawn, at least not without a caveat: the respondents in this study were given the information and were incentivized to read it (they were paid respondents). In the real world, it's not clear how many voters will actually read the CIR recommendation, unprompted. If the statement goes unread, the gains go unrealized. This should be mentioned somewhere.

Methods/analysis/ results

- P. 6 “About 6 percent more of respondents pick the correct answer on individual knowledge items…” Is this difference significant? Should be reported in the text.

- Also, please present the standard error of the mean difference in parentheses in the body of the text.

- P.8 “An additional three cases fit this pattern at a lower level of statistical significance (p .10).” No, this is not how p-values work. Your alpha (level of significance) is an arbitrary threshold decided in advance. Unless an alpha of 0.10 was specified in advance, just use the default of 0.05. Which means any p-value greater than 0.05 is simply not significant.

Grammar/Style

- Define acronyms the first time they appear (i.e., Line 3 p. 45 RAND corporation)

- P. 4 line 77/78 there’s a problem with this sentence: "thought one mail survey used a registered voter list and another used a 78 registered voter list."

But in general, an excellent paper that will make an excellent contribution to the discipline.

6. PLOS authors have the option to publish the peer review history of their article (what does this mean?). If published, this will include your full peer review and any attached files.

Reviewer #1: **Yes: **Mark E Warren

Reviewer #2: **Yes: **Edana Beauvais

---

## [Author Response · Author response to Decision Letter 0]

8 Jun 2023

The reply to reviewers is an attached file, which includes screenshots of edits that would not appear in this text-only field.

---

## [Editor Report · Decision Letter 1]

21 Jun 2023

Deliberative panels as a source of public knowledge: A large-sample test of the Citizens’ Initiative Review

PONE-D-23-06764R1

Dear Dr. Gastil,

We’re pleased to inform you that your manuscript has been judged scientifically suitable for publication and will be formally accepted for publication once it meets all outstanding technical requirements.

Kind regards,

Jean-François Daoust

Academic Editor

PLOS ONE

Additional Editor Comments (optional):

The set of reviews were helpful and constructive; you were responsive and engaged with them in a satisfying manner. This manuscript is improved and meets, in my view, PLOS One's criteria for publication. Congratulations.
---

## [Editor Report · Acceptance letter]

19 Jul 2023

PONE-D-23-06764R1 

Deliberative panels as a source of public knowledge: A large-sample test of the Citizens’ Initiative Review 

Dear Dr. Gastil:

I'm pleased to inform you that your manuscript has been deemed suitable for publication in PLOS ONE. Congratulations! Your manuscript is now with our production department. 

Kind regards, 

on behalf of

Dr. Jean-François Daoust 

Academic Editor

PLOS ONE